# Understanding Architectures Learnt by Cell-based Neural Architecture Search

**Yao Shu, Wei Wang & Shaofeng Cai**
School of Computing
National University of Singapore
{shuyao,wangwei,shaofeng}@comp.nus.edu.sg

## Abstract

Neural architecture search (NAS) searches architectures automatically for given tasks, e.g., image classification and language modeling. Improving the search efficiency and effectiveness has attracted increasing attention in recent years. However, few efforts have been devoted to understanding the generated architectures. In this paper, we first reveal that existing NAS algorithms (e.g., DARTS, ENAS) tend to favor architectures with wide and shallow cell structures. These favorable architectures consistently achieve fast convergence and are consequently selected by NAS algorithms. Our empirical and theoretical study further confirms that their fast convergence derives from their smooth loss landscape and accurate gradient information. Nonetheless, these architectures may not necessarily lead to better generalization performance compared with other candidate architectures in the same search space, and therefore further improvement is possible by revising existing NAS algorithms.

## 1 Introduction

Various neural network architectures (Krizhevsky et al., 2012; Simonyan & Zisserman, 2015; He et al., 2016; Huang et al., 2017) have been devised over the past decades, achieving superhuman performance for a wide range of tasks. Designing these neural networks typically takes substantial efforts from domain experts by trial and error. Recently, there is a growing interest in neural architecture search (NAS), which automatically searches for high-performance architectures for the given task. The searched NAS architectures (Zoph et al., 2018; Real et al., 2019; Pham et al., 2018; Liu et al., 2019; Xie et al., 2019b; Luo et al., 2018; Cai et al., 2019; Akimoto et al., 2019; Nayman et al., 2019) have outperformed best expert-designed architectures on many computer vision and natural language processing tasks.

Mainstream NAS algorithms typically search for the connection topology and transforming operation accompanying each connection from a predefined search space. Tremendous efforts have been exerted to develop efficient and effective NAS algorithms (Liu et al., 2019; Xie et al., 2019b; Luo et al., 2018; Akimoto et al., 2019; Nayman et al., 2019). However, less attention has been paid to these searched architectures for further insight. To our best knowledge, there is no related work in the literature examining whether these NAS architectures share any pattern, and how the pattern may impact the architecture search if there exists the pattern. These questions are fundamental to understand and improve existing NAS algorithms. In this paper, we endeavour to address these questions by examining the popular NAS architectures[1].

The recent work (Xie et al., 2019a) shows that the architectures with random connection topologies can achieve competitive performance on various tasks compared with expert-designed architectures. Inspired by this result, we examine the connection topologies of the architectures generated by popular NAS algorithms. In particular, we find a connection pattern of the popular NAS architectures. These architectures tend to favor wide and shallow cells, where the majority of intermediate nodes are directly connected to the input nodes.

---

[1]The popular NAS architectures refer to the best architectures generated by state-of-the-art (SOTA) NAS algorithms throughout the paper. Notably, we research on the cell-based NAS algorithms and architectures.

To appreciate this particular connection pattern, we first visualize the training process of the popular NAS architectures and their randomly connected variants. Fast and stable convergence is observed for the architectures with wide and shallow cells. We further empirically and theoretically show that the architectures with wider and shallower cells consistently enjoy a smoother loss landscape and smaller gradient variance than their random variants, which helps explain their better convergence and consequently the selection of these NAS architectures during the architecture search.

We finally evaluate the generalization performance of the popular NAS architectures and their randomly connected variants. We find that the architectures with wide and shallow cells may not generalize better than other candidate architectures despite their faster convergence. We therefore believe that rethinking NAS from the perspective of the true generalization performance rather than the convergence of candidate architectures should potentially help generate better architectures.

## 2    RELATED WORKS

**Neural Architecture Search**    Neural architecture search (NAS) searches for best-performing architectures automatically for a given task. It has received increasing attention in recent years due to its outstanding performance and the demand for automated machine learning (AutoML). There are three major components in NAS as summarized by Elsken et al. (2019), namely search space, search policy (or strategy, algorithm), and performance evaluation (or estimation). To define the search space, the prior knowledge extracted from expert-designed architectures is typically exploited. As for the search policy, different algorithms are proposed to improve the effectiveness (Zoph et al., 2018; Real et al., 2019; Tan et al., 2019; Cai et al., 2019) and the efficiency (Pham et al., 2018; Liu et al., 2019; Xie et al., 2019b; Luo et al., 2018; Nayman et al., 2019; Akimoto et al., 2019) of the architecture search. However, no effort has been devoted to understanding the best architectures generated by various NAS approaches. Detailed analysis of these architectures may give insights about the further improvement of existing NAS algorithms.

**Evaluation of NAS algorithms**    Recent works evaluate NAS algorithms by comparing them with random search. Li & Talwalkar (2019) and Sciuto et al. (2019) compare the generalization performance of architectures generated from random search and existing NAS algorithms. Interestingly, the random search can find architectures with comparable or even better generalization performance. Particularly, Sciuto et al. (2019) show empirically that the ineffectiveness of some NAS algorithms (Pham et al., 2018) could be the consequence of the weight sharing mechanism during the architecture search. While these evaluations help understand the general disadvantages of NAS algorithms, what kind of architectures the NAS algorithms are learning and why they learn these specific architectures are still not well understood.

## 3    THE CONNECTION PATTERN OF POPULAR NAS CELLS

Mainstream NAS algorithms (Zoph et al., 2018; Real et al., 2019; Pham et al., 2018; Liu et al., 2019; Xie et al., 2019b; Luo et al., 2018) typically search for the cell structure, including the connection topology and the corresponding operation (transformation) coupling each connection. The generated cell is then replicated to construct the entire neural network. We therefore mainly investigate these cell-based NAS architectures. In this section, we first introduce the commonly adopted cell representation, which is useful to understand the connection and computation in a cell space. We then sketch the connection topologies of popular cell-based NAS architectures to investigate their connection patterns. By comparison, we show that there is a common connection pattern among the cells learned by different NAS algorithms; particularly, these cells tend to be wide and shallow.

### 3.1    CELL REPRESENTATION

Following DARTS (Liu et al., 2019), we represent the cell topology as a directed acyclic graph (DAG) consisting of $N$ nodes, including $M$ input nodes, one output node and $(N - M - 1)$ intermediate nodes. Each node forms a latent representation of the input instance. The input nodes consist of the outputs from $M$ preceding cells. And the output node aggregates (e.g., concatenate) the representations from all intermediate nodes. Each intermediate node is connected to $M$ proceeding nodes in the same cell. Each connection transforms the representation from one node via

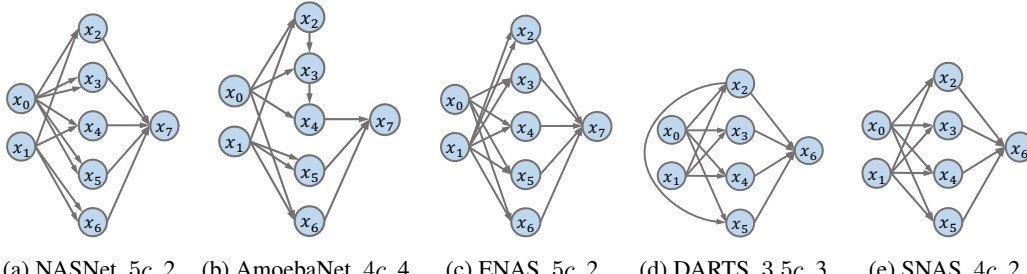

(a) NASNet, $5c$, 2    (b) AmoebaNet, $4c$, 4    (c) ENAS, $5c$, 2    (d) DARTS, $3.5c$, 3    (e) SNAS, $4c$, 2

Figure 1: Cell topologies of popular NAS architectures. Each sub-figure has three sets of nodes from left to right, i.e., the input nodes, intermediate nodes, and output node. The arrows (i.e., operations of the cell) represent the direction of information flow. The caption of each sub-figure reports the name of the architecture, width and depth of a cell following our definition. The width of a cell is computed with the assumption that all intermediate nodes share the same width $c$.

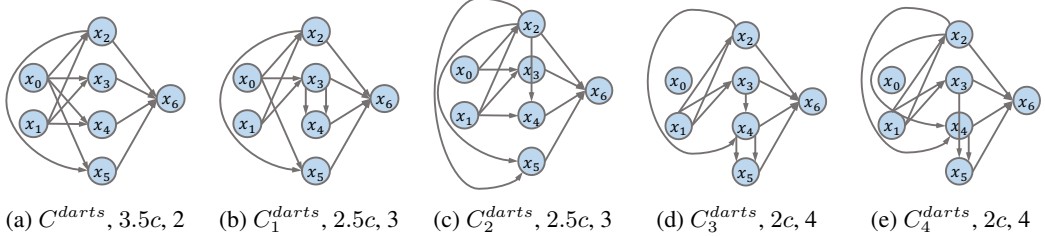

(a) $C^{darts}$, $3.5c$, 2    (b) $C_1^{darts}$, $2.5c$, 3    (c) $C_2^{darts}$, $2.5c$, 3    (d) $C_3^{darts}$, $2c$, 4    (e) $C_4^{darts}$, $2c$, 4

Figure 2: Topologies of DARTS (Liu et al., 2019) cell (leftmost) and its variants with random connections. The cell depth is increasing and width decreasing from left to right. In particular, the original DARTS cell $C^{darts}$ is widest and shallowest among these cells.

an operation from a predefined operation set, e.g., 3×3 convolution, 3×3 max pooling, etc. The target of NAS algorithm is to search for the best $M$ source nodes for each intermediate node and the best operation for each of the connections between nodes. In the literature, the searched cell is then replicated by $L$ times to build the entire neural network architecture[2].

We abuse the notation $C$ to denote a cell and also the architecture built with the specific cell in the following sections. Besides, we shall use $C^A$ to denote the best architecture (or cell) searched with the NAS algorithm $A$ (e.g., DARTS (Liu et al., 2019), ENAS (Pham et al., 2018)). Details on how to build the architecture with given cells are provided in Appendix A.3.

## 3.2 THE COMMON CONNECTION PATTERN

Recently, Xie et al. (2019a) shows that neural networks constructed by cells with random connection patterns can achieve compelling performance on multiple tasks. Taking this a step further, we wonder whether cells generated from popular NAS algorithms share any connection patterns, which may explain why these cells are chosen during the architecture search. To investigate the connection patterns, we sketch the topologies of the popular NAS cells with detailed operations omitted.

Figure 1 illustrates topologies of 5 popular NAS cells[3]. To examine the connection pattern formally, we introduce the concept of 'depth' and 'width' for a cell. The depth of a cell is defined as the number of connections along the longest path from input nodes to the output node. The width of a cell is defined as the total width of the intermediate nodes that are connected to the input nodes. In particular, if some intermediate nodes are only partially connected to input nodes (i.e., have connections to other intermediate nodes), their width is reduced by the percentage of the number of connections to intermediate nodes over all connections. The width of a node is the number

---

[2]We omit reduction cell here for brevity, which is used for dimension reduction in NAS.

[3]We only visualize normal cells since the number of normal cells is significantly larger than the reduction cells in popular NAS architectures.

of channels for convolution operations; and the width is the dimension of the features for linear operations. Supposing that the width of each intermediate node is $c$, as shown in Figure 1, the width and depth of the DARTS (Liu et al., 2019) cell are $3.5c$ and 3 respectively, and the width and depth of the AmoebaNet (Real et al., 2019) cell are $4c$ and 4 correspondingly.

Following the above definitions, the smallest depth and largest width for a cell with $N = 7$ and $M = 2$ are 2 and $4c$ respectively. Similarly, for a cell with $N = 8$ and $M = 2$, the smallest depth and largest width are 2 and $5c$ respectively. In Figure 1, we can observe that cells from popular NAS architectures tend to be the widest and shallowest ones (with width close to $4c/5c$ and depth close to 2) among all candidate cells in the same search space. Regarding this as the common connection pattern, we have the following observation:

**Observation 3.1 (The Common Connection Pattern)** *NAS architectures generated by popular NAS algorithms tend to have the widest and shallowest cells among all candidate cells in the same search space.*

## 4    THE IMPACTS OF CELL WIDTH AND DEPTH ON OPTIMIZATION

Given that popular NAS cells share the common connection pattern, we then explore the impact of this common connection pattern from the optimization perspective to answer the question: *why the wide and shallow cells are selected during the architecture search?* We sample and train variants of popular NAS architectures with random connections. Comparing randomly connected variants with the popular NAS architectures, we find that architectures with wider and shallower cells indeed converge faster so that they are selected by NAS algorithms (Section 4.1). To understand why the wider and shallower cell contributes to faster convergence, we further investigate the loss landscape and gradient variance of popular NAS architectures and their variants via both empirical experiments (Section 4.2) and theoretical analysis (Section 4.3).

### 4.1    CONVERGENCE

Popular NAS algorithms typically evaluate the performance of a candidate architecture prematurely before the convergence of its model parameters during the search process. For instance, DARTS (Liu et al., 2019), SNAS (Xie et al., 2019b) and ENAS (Pham et al., 2018) optimize hyper-parameters of architectures and model parameters concurrently. The amortized training time of each candidate architecture is insufficient and therefore far from the requirement for the full convergence. Likewise, AmoebaNet (Real et al., 2019) evaluates the performance of candidate architectures with the training of only a few epochs. In other words, these candidate architectures are not evaluated based on their generalization performance at convergence. As a result, architectures with faster convergence rates are more likely to be selected by existing NAS algorithms because they can obtain better evaluation performance given the same training budget. We therefore hypothesize that the popular NAS architectures may converge faster than other candidate architectures, which largely contributes to the selection of these architectures during the search.

To support the hypothesis above, we compare the convergence of original NAS architectures and their variants with random connections via empirical studies. We first sample variants of popular NAS cells following the sampling method in Appendix A.2. Then, we train both original NAS architectures and their random variants on CIFAR-10 and CIFAR-100 following the training details in Appendix A.3. During training, we evaluate the testing loss and accuracy of these architectures. Since the convergence is dependent on optimization settings, we also evaluate the convergence performance under different learning rates.

Take DARTS (Liu et al., 2019) for example, Figure 2 shows the connection topology of the original DARTS cell and its random variants. Figure 3 reports the test loss and accuracy curves of these architectures during training. As illustrated in Figure 3, the original cell $C^{darts}$, known as the widest and shallowest cell, has the fastest and most stable convergence compared with its variants. Further, as the width of a cell increases and the depth decreases (i.e., from $C_4$ to $C_1$), the convergence becomes faster. The results of other popular NAS architectures and their randomly connected variants are reported in Appendix B.2.

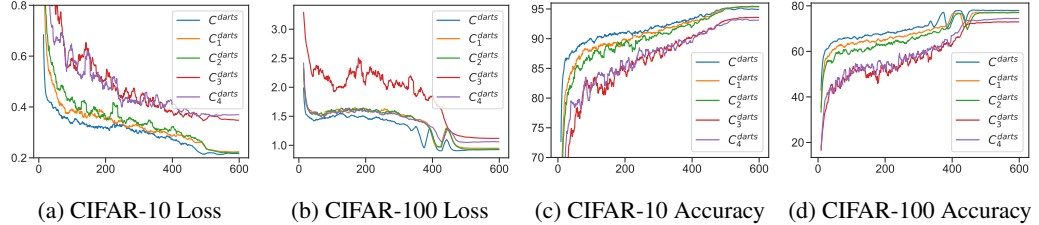

| (a) CIFAR-10 Loss | (b) CIFAR-100 Loss | (c) CIFAR-10 Accuracy | (d) CIFAR-100 Accuracy |

Figure 3: Test loss and test accuracy (%) curves of DARTS and its randomly connected variants on CIFAR-10 and CIFAR-100 during training. The default learning rate is 0.025.

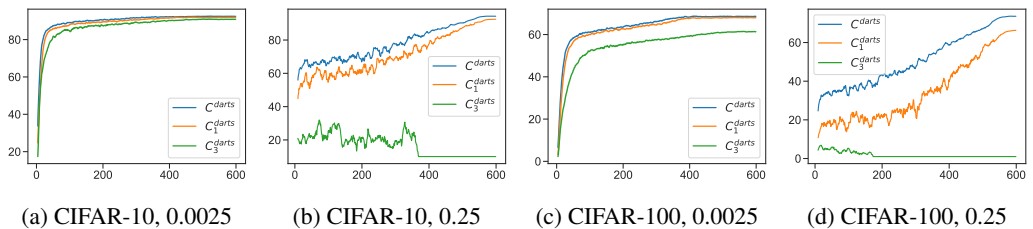

| (a) CIFAR-10, 0.0025 | (b) CIFAR-10, 0.25 | (c) CIFAR-100, 0.0025 | (d) CIFAR-100, 0.25 |

Figure 4: Test accuracy (%) curves of DARTS and its randomly connected variants on CIFAR-10 and CIFAR-100 during training under different learning rates (0.0025 and 0.25). We only evaluate $C^{darts}$, $C_1^{darts}$ and $C_3^{darts}$ for illustration. The caption of each sub-figure reports the dataset and the learning rate.

Figure 4 further validates the difference of convergence under different learning rates. The original cell $C^{darts}$ enjoys the fastest and the most stable convergence among these cells under various learning rates. The difference in terms of convergence rate and stability is more obvious between $C^{darts}$ and its variants with a larger learning rate as shown in Figure 4. Interestingly, $C_3^{darts}$ completely fails to converge on both CIFAR-10 and CIFAR-100 with a larger learning rate of 0.25. While there is a minor difference among these cells with a lower learning rate of 0.0025, we still find that there is a decreasing performance of convergence (i.e., convergence rate and stability) from $C^{darts}$, $C_1^{darts}$ to $C_3^{darts}$. Overall, the observations are consistent with the results in Figure 3.

We have also compared the convergence of popular NAS architectures and their random variants of different operations. Similarly, we sample and train the random variants of operations for popular NAS architectures following the details in Appendix A.2 and Appendix A.3. Figure 5 illustrates the convergence of these architectures. Surprisingly, with the same connection topologies as the popular NAS cells but different operations, all random variants achieve nearly the same convergence as these popular NAS architectures. Consistent results can be found in Figure 12 of Appendix B.2. We therefore believe that the types of operations have limited impacts on the convergence of NAS architectures and the connection topologies affect the convergence more significantly.

With these observations, we conclude that the popular NAS architectures with wider and shallower cells indeed converge faster and more stably, which explains why these popular NAS cells are selected during the architecture search. The next question is then *why the wider and shallower cell leads to a faster and more stable convergence?*

## 4.2 EMPIRICAL STUDY OF FACTORS AFFECTING CONVERGENCE

Since the wide and shallow cell is related to fast convergence, we further conduct the theoretical convergence analysis to investigate the cause of fast convergence. In this section, we first introduce the convergence analysis (i.e., Theorem 4.1) of non-convex optimization with the randomized stochastic gradient method (Ghadimi & Lan, 2013). Based on the analysis, we introduce the possible factors related to the common connection pattern that may affect the convergence. We then examine these factors empirically in the following subsections.

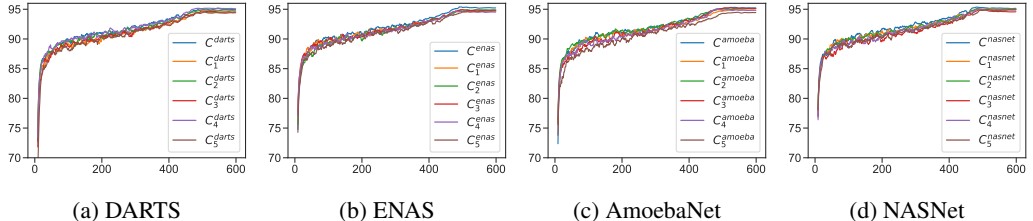

(a) DARTS     (b) ENAS     (c) AmoebaNet     (d) NASNet

Figure 5: Test accuracy (%) curves of DARTS, ENAS, AmoebaNet, NASNet and their random variants of operations on CIFAR-10 during training. The parameter size is attached in Table 3 of Appendix B.2.

**Theorem 4.1** *(Ghadimi & Lan, 2013) Let $f$ be a $L$-smooth non-convex function, and let $f^*$ be the minimal. Given repeated, independent accesses to stochastic gradients with variance bound $\sigma^2$ for $f(\boldsymbol{w})$, SGD with initial $\boldsymbol{w}_0$, total iterations $N > 0$ and learning rate $\gamma_k < \frac{1}{L}$ achieves the following convergence by randomly choosing $\boldsymbol{w}_k$ as the final output $\boldsymbol{w}_R$ with probability $\frac{\gamma_k}{H}$ where $H = \sum_{k=1}^{N} \gamma_k$:*

$$\mathbb{E}[\|\nabla f(\boldsymbol{w}_R)\|^2] \leq \frac{2(f(\boldsymbol{w}_0) - f^*)}{H} + \frac{L\sigma^2}{H} \sum_{k=1}^{N} \gamma_k^2$$

In this paper, $f$ and $\boldsymbol{w}$ denote the objective (loss) function and model parameters respectively. Based on the above theorem, Lipschitz smoothness $L$ and gradient variance $\sigma^2$ significantly affect the convergence, including the rate and the stability of convergence. Particularly, given a specific number of iterations $N$, a smaller Lipschitz constant $L$ or smaller gradient variance $\sigma^2$ would lead to a smaller convergence error and less damped oscillations, which indicates a faster and more stable convergence. Since the Lipschitz constant $L$ and gradient variance $\sigma^2$ are highly related to the objective function, different NAS architectures result in different $L$ and $\sigma^2$. In the following subsections, we therefore conduct empirical analysis for the impacts of the cell with and depth on the Lipschitz smoothness and gradient variance.

### 4.2.1 Loss Landscape

The constant $L$ of Lipschitz smoothness is closely correlated with the Hessian matrix of the objective function as shown by Nesterov (2004), which requires substantial computation and can only represent the global smoothness. The loss contour, which has been widely adopted to visualize the loss landscape of neural networks by Goodfellow & Vinyals (2015); Li et al. (2018), is instead computationally efficient and is able to report the local smoothness of the objective function. To explore the loss landscape of different architectures, we adopt the method in Li et al. (2018) to plot the loss contour $s(\alpha, \beta) = \mathbb{E}_{i \sim P}[f_i(\boldsymbol{w}^* + \alpha \boldsymbol{w}_1 + \beta \boldsymbol{w}_2)]$. The notation $f_i(\cdot)$ denotes the loss evaluated at $i_{th}$ instance in the dataset and $P$ denotes the distribution of dataset. The notation $\boldsymbol{w}^*$, $\boldsymbol{w}_1$ and $\boldsymbol{w}_2$ denote the (local) optimal and two direction vectors randomly sampled from Gaussian distribution respectively. And $\alpha$, $\beta$, which are the $x$ and $y$ axis of the plots, denote the step sizes to perturb $\boldsymbol{w}^*$. The loss contour plotted here is therefore a two-dimensional approximation of the truly high-dimensional loss contour. However, as shown in Li et al. (2018), the approximation is valid and effective to characterize the property of the true loss contour.

To study the impact of the cell width and depth on Lipschitz smoothness, we compare the loss landscape between popular NAS architectures and their randomly connected variants trained in Section 4.1 on CIFAR-10 and CIFAR-100. Due to the space limitation, we only plot the loss landscape of DARTS (Liu et al., 2019) and its randomly connected variants in Figure 6. We observe that the connection topology has a significant influence on the smoothness of the loss landscape. With the widest and shallowest cell, $C^{darts}$ has a fairly benign and smooth landscape along with the widest near-convex region around the optimal. With a deeper and narrower cell, $C_1^{darts}$ and $C_2^{darts}$ have a more agitated loss landscape compared with $C^{darts}$. Further, $C_3^{darts}$, with the smallest width and largest depth among these cells, has the most complicated loss landscape and the narrowest and steepest near-convex region around the optimum. The largest eigenvalue of the Hessian matrix, which indicates the maximum curvature of the objective function, is positively correlated with Lip-

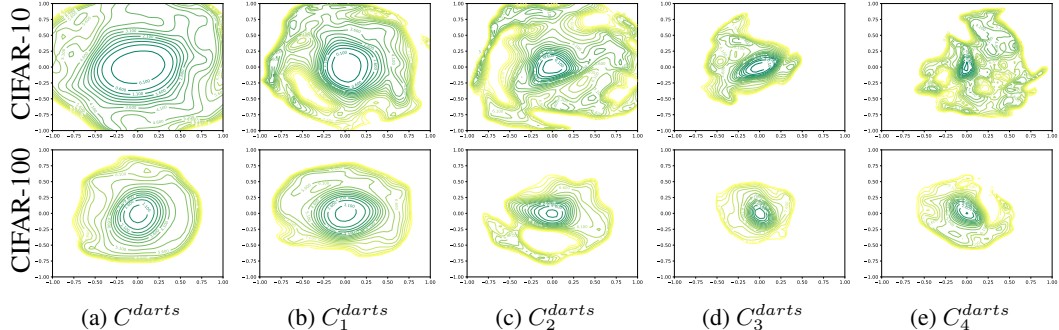

Figure 6: Loss contours of DARTS and its variants with random connections on the test dataset of CIFAR-10. The lighter color of the contour lines indicates a larger loss. Notably, the loss of the blank area, around the corners of each plot, is extremely large. Besides, the area with denser contour lines indicates a steeper loss surface.

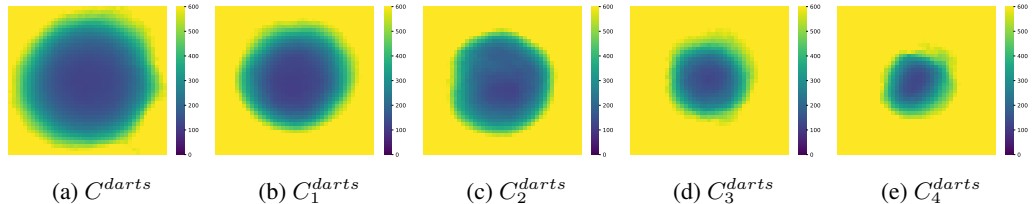

Figure 7: Heat maps of the gradient variance from DARTS and its randomly connected variants around the optimal on the test dataset of CIFAR-10. The lighter color indicates a larger gradient variance. Notably, the gradient variance of the yellow area, around the corners of each plot, is extremely large. Obviously, the region with relatively small gradient variance becomes smaller from left to right.

schitz constant as shown by Nesterov (2004). A smoother loss landscape therefore corresponds to a smaller Lipschitz constant $L$. $C^{darts}$ is likely to achieve the smallest Lipschitz constant among these cells.

Consistent results can be found in Appendix B.3 for the loss landscape of other popular NAS cells and their variants. Based on these results, we conclude that increasing the width and decreasing the depth of a cell widens the near-convex region around the optimal and smooths the loss landscape. The constant $L$ of Lipschitz smoothness therefore becomes smaller locally and globally. Following Theorem 4.1, architectures with wider and shallower cells shall converge faster and more stably.

### 4.2.2 GRADIENT VARIANCE

The gradient variance indicates the noise level of gradient by randomly selecting training instances in stochastic gradient descent (SGD) method. Large gradient variance indicates large noise in the gradient, which typically results in unstable updating of model parameters. Following Ghadimi & Lan (2013), gradient variance is defined as $\mathrm{Var}(\nabla f_i(\boldsymbol{w}))$. Similar to the visualization of loss landscape in Section 4.2.1, we visualize the gradient variance by $g(\alpha, \beta) = \mathrm{Var}(\nabla f_i(\boldsymbol{w}^* + \alpha \boldsymbol{w}_1 + \beta \boldsymbol{w}_2))$. All other notations follow Section 4.2.1.

To study the impact of the width and depth of a cell on the gradient variance, we compare the gradient variance between popular NAS architectures and their randomly connected variants trained in Section 4.1 on CIFAR-10 and CIFAR-100. We visualize the gradient variance of DARTS (Liu et al., 2019) and its randomly connected variants in Figure 7 and Figure 8. For better visualization, we plot the figures using the standard deviation (i.e., $\sqrt{g(\alpha, \beta)}$) to avoid extremely large values in the visualization of DARTS. Obviously, as the cell width decreases and the cell depth increases (i.e., from $C^{darts}$ to $C_4^{darts}$), the region with relatively small gradient variance becomes smaller as shown in Figure 7. Consistently, the gradient variance generally shows an increasing trend from $C^{darts}$ to

(a) $C^{darts}$     (b) $C_1^{darts}$     (c) $C_2^{darts}$     (d) $C_3^{darts}$     (e) $C_4^{darts}$

Figure 8: 3D surfaces of the gradient variance from DARTS and its randomly connected variants around the optimal on the test dataset of CIFAR-100. The height of the surface indicates the value of gradient variance. Notably, the height of the gradient variance surface is gradually increasing from left to right. Especially, $C^{darts}$ has the smoothest and lowest surface of gradient variance among these architectures.

$C_4^{darts}$ in Figure 8. Consequently, the gradient becomes noisier in the neighborhood of the optimal, which typically makes the optimization harder and unstable.

Similar results from other popular NAS architectures and their random variants are provided in Appendix B.4. Based on these results, we conclude that the increase in width and the decrease in depth of a cell result in a smaller gradient variance, which makes the optimization process less noisy and more efficient. The convergence of wide and shallow cells therefore shall be fast and stable following Theorem 4.1.

### 4.3 THEORETICAL ANALYSIS OF FACTORS AFFECTING CONVERGENCE

Our empirical study so far suggests that larger cell width and smaller cell depth smooth the loss landscape and decrease the gradient variance. Consequently, popular NAS architectures with wide and shallow cells converge fast. In this section, we investigate the impacts of the cell width and depth on Lipschitz smoothness and gradient variance from a theoretical perspective.

#### 4.3.1 SETUP

We analyze the impact of the cell width and depth by comparing architectures with the widest cell and the narrowest cell as shown in Figure 26 of Appendix C. To simplify the analysis, the cells we investigate contain only one input node $x$ and one output node. The input node may be training instances or output node from any proceeding cell. All operations in the cell are linear operations without any non-linearity. Suppose there are $n$ intermediate nodes in a cell, the $i_{th}$ intermediate node and its associated weight matrix are denoted as $y^{(i)}$ and $W^{(i)}(i = 1, \cdots, n)$ respectively. The output node $z$ denotes the concatenation of all intermediate nodes. Both cells have the same arbitrary objective function $f$ following the output node, which shall consist of the arbitrary number of activation functions and cells. For clarity, we refer to the objective function, intermediate nodes and output node of the architecture with the narrowest cell as $\widehat{f}$, $\widehat{y}^{(i)}$ and $\widehat{z}$ respectively. As shown in Figure 26, the intermediate node $y^{(i)}$ and $\widehat{y}^{(i)}$ can be computed by $y^{(i)} = W^{(i)}x$ and $\widehat{y}^{(i)} = \prod_{k=1}^{i} W^{(k)}x$ respectively. Particularly, we set $\prod_{k=1}^{i} W^{(k)} = W^{(i)}W^{(i-1)} \cdots W^{(1)}$. And all the related proofs of following theorems can be found in Appendix C.

#### 4.3.2 THEORETICAL RESULTS

Due to the complexity of the standard Lipschitz smoothness, we instead investigate the block-wise Lipschitz smoothness (Beck & Tetruashvili, 2013) of the two cases shown in Figure 26. In Theorem 4.2, we show that the block-wise Lipschitz constant of the narrowest cell is scaled by the largest eigenvalues of the model parameters (i.e., $W^{(i)}(i = 1, \cdots, n)$). Notably, the Lipschitz constant of the narrowest cell can be significantly larger than the one of the widest cell while most of the largest eigenvalues are larger than 1, which slows down the convergence substantially. The empirical study in Section 4.2.1 has validated the results.

**Theorem 4.2 (The impact of cell width and depth on block-wise Lipschitz smoothness )** *Let* $\lambda^{(i)}$ *be the largest eigenvalue of* $W^{(i)}$. *Given the widest cell with objective function* $f$ *and the*

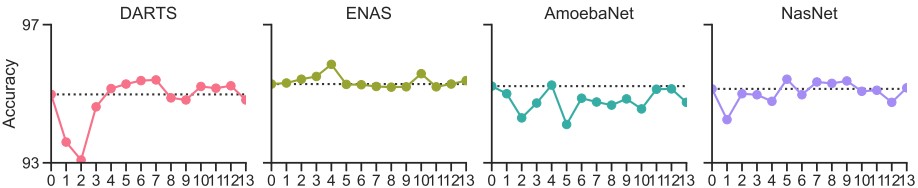

Figure 9: Comparison of the test accuracy at the convergence between popular NAS architectures and their randomly connected variants on CIFAR-10. Each popular NAS architecture (index 0 on the $x$-axis) is followed by 13 randomly connected variants (from index 1 to index 13 on the $x$-axis), corresponding to $C_1$ to $C_{13}$ respectively. The width and depth of these random variants are shown in Table 2 in Appendix B.2. The dashed lines report the accuracy of the popular NAS architectures.

narrowest cell with objective function $\widehat{f}$, by assuming the block-wise Lipschitz smoothness of the widest cell as $\left\|\frac{\partial f}{\partial W_1^{(i)}} - \frac{\partial f}{\partial W_2^{(i)}}\right\| \leq L^{(i)} \left\|W_1^{(i)} - W_2^{(i)}\right\|$ for any $W_1^{(i)}$ and $W_2^{(i)}$, the block-wise Lipschitz smoothness of the narrowest cell then can be represented as

$$\left\|\frac{\partial \widehat{f}}{\partial W_1^{(i)}} - \frac{\partial \widehat{f}}{\partial W_2^{(i)}}\right\| \leq (\prod_{j=1}^{i-1} \lambda^{(j)}) L^{(i)} \left\|W_1^{(i)} - W_2^{(i)}\right\|$$

We then compare the gradient variance of the two cases shown in Figure 26. Interestingly, gradient variance suggests a similar but more significant difference between the two cases compared with their difference in Lipschitz smoothness. As shown in Theorem 4.3, the gradient variance of the narrowest cell is not only scaled by the square of the largest eigenvalue of the weight matrix but also is scaled by the number of intermediate nodes (i.e., $n$). Moreover, the upper bound of its gradient variance has numbers of additional terms, leading to a significantly larger gradient variance. The empirical study in Section 4.2.2 has confirmed the results.

**Theorem 4.3 (The impact of cell width and depth on gradient variance )** *Let $\lambda^{(i)}$ be the largest eigenvalue of $W^{(i)}$. Given the widest cell with objective function $f$ and the narrowest cell with objective function $\widehat{f}$, by assuming the gradient variance of the widest cell as $\mathbb{E}\left\|\frac{\partial f}{\partial W^{(i)}} - \mathbb{E}\frac{\partial f}{\partial W^{(i)}}\right\|^2 \leq (\sigma^{(i)})^2$ for any $W^{(i)}$, the gradient variance of the narrowest cell is then bounded by*

$$\mathbb{E}\left\|\frac{\partial \widehat{f}}{\partial W^{(i)}} - \mathbb{E}\frac{\partial \widehat{f}}{\partial W^{(i)}}\right\|^2 \leq n \sum_{k=i}^{n} (\frac{\sigma^{(k)}}{\lambda^{(i)}} \prod_{j=1}^{k} \lambda^{(j)})^2$$

## 5 GENERALIZATION BEYOND THE COMMON CONNECTIONS

Our empirical and theoretical results so far have demonstrated that the common connection pattern helps to smooth the loss landscape and make the gradient more accurate. Popular NAS architectures with wider and shallower cells therefore converge faster, which explains why popular NAS architectures are selected by the NAS algorithms. Nonetheless, we have ignored the generalization performance obtained by popular NAS architectures and their random variants. We therefore wonder *whether popular NAS architectures with wide and shallow cells generalize better*.

In Figure 9, we visualize the test accuracy of popular NAS architectures and their randomly connected variants trained in Section 4.1. Notably, the popular NAS architectures can achieve competitive accuracy compared with most of the random variants. However, there are some random variants, which achieve higher accuracy than the popular architectures. Interestingly, there seems to be an optimal choice of depth and width for a cell to achieve higher test accuracy (i.e., $C_7$ for DARTS and $C_4$ for ENAS). Popular NAS architectures with wide and shallow cells therefore are not guaranteed to generalize better, although they typically converge faster than other random variants.

We also adapt the connections of popular NAS architectures to obtain their widest and shallowest variants. The adaption is possible due to the fact that the cells (including normal and reduction cell)

Table 1: Comparison of the test error at the convergence between the original and the adapted NAS architectures on CIFAR-10/100 and Tiny-ImageNet-200. The entire networks are constructed and trained following the experimental settings reported in Appendix A.3, which may slightly deviate from the original ones. The test errors (or the parameter sizes) of original and adapted architectures are reported on the left and right hand-side of slash respectively.

| Architecture | CIFAR-10 | | CIFAR-100 | | Tiny-ImageNet-200 | |
|---|---|---|---|---|---|---|
| | **Error**(%) | **Params**(M) | **Error**(%) | **Params**(M) | **Error**(%) | **Params**(M) |
| NASNet (Zoph et al., 2018) | **2.65**/2.80 | 4.29/4.32 | 17.06/**16.86** | 4.42/4.45 | **31.88**/32.05 | 4.57/4.60 |
| AmoebaNet (Real et al., 2019) | **2.76**/2.91 | 3.60/3.60 | 17.55/**17.28** | 3.71/3.71 | **32.22**/33.16 | 3.83/3.83 |
| ENAS (Pham et al., 2018) | **2.64**/2.76 | 4.32/4.32 | 16.67/**16.04** | 4.45/4.45 | **30.68**/31.36 | 4.60/4.60 |
| DARTS (Liu et al., 2019) | **2.67**/2.73 | 3.83/3.90 | 16.41/**16.15** | 3.95/4.03 | **30.58**/31.33 | 4.08/4.16 |
| SNAS (Xie et al., 2019b) | 2.88/**2.69** | 3.14/3.19 | 17.78/**17.20** | 3.26/3.31 | **32.40**/32.61 | 3.39/3.45 |

of popular NAS architectures are generally not widest and narrowest as shown in Figure 1. While there are various widest and shallowest cells following our definition of cell width and depth, we apply the connection pattern of SNAS cell shown in Figure 1(e) to obtain the widest and shallowest cells. The adapted topologies are shown in Figure 25 of Appendix B.5.

Table 1 illustrates the comparison of the test accuracy between our adapted NAS architectures and the original ones. As shown in Table 1, the adapted architectures achieve smaller test error on CIFAR-100. Nevertheless, most of the adapted architectures, obtain larger test error than the original NAS architectures on both CIFAR-10 and Tiny-ImageNet-200[4]. The results again suggest that the widest and shallowest cells may not help architectures generalize better, while these architectures typically achieve compelling generalization performance.

The results above have revealed that the architectures with wide and shallow cells may not generalize better despite their fast convergence. To improve current NAS algorithms, we therefore need to rethink the evaluation of the performance of candidate architectures during architecture search since the current NAS algorithms are not based on the generalization performance at convergence as mentioned in Section 4.1. Nonetheless, architectures with the wide and shallow cells usually guarantee a stable and fast convergence along with competitive generalization performance, which should be good prior knowledge for designing architectures and NAS algorithms.

# 6 CONCLUSION AND DISCUSSION

Recent works have been focusing on the design and evaluation of NAS algorithms. We instead endeavour to examine the architectures selected by the various popular NAS algorithms. Our study is the first to explore the common structural patterns selected by existing algorithms, why these architectures are selected, and why these algorithms may be flawed. In particular, we reveal that popular NAS algorithms tend to favor architectures with wide and shallow cells, which typically converge fast and consequently are likely be selected during the search process. However, these architectures may not generalize better than other candidates of narrow and deep cells.

To further improve the performance of the selected NAS architectures, one promising direction for the current NAS research is to evaluate the generalization performance of candidate architectures more accurately and effectively. While popular NAS architectures appreciate fast and stable convergence along with competitive generalization performance, we believe that the wide and shallow cells are still useful prior knowledge for the design of the search space. We hope this work can attract more attention to the interpretation and understanding of existing popular NAS algorithms.

ACKNOWLEDGEMENT

This research is supported by the National Research Foundation Singapore under its AI Singapore Programme [Award No. AISG-GC-2019-002] and Singapore Ministry of Education Academic Research Fund Tier 3 under MOEs official grant number MOE2017-T3-1-007.

---

[4]https://tiny-imagenet.herokuapp.com/

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

## APPENDIX A    EXPERIMENTAL SETUP

### A.1    DATA PRE-PROCESSING AND AUGMENTATION

Our experiments are conducted on CIFAR-10/100 (Krizhevsky et al., 2009) and Tiny-ImageNet-200. CIFAR-10/100 contains 50,000 training images and 10,000 test images of $32 \times 32$ pixels in 10 and 100 classes respectively. Tiny-ImageNet-200 consists of 100,000 training images, 10,000 validation images and 10,000 test images[5] in 200 classes. We adopt the same data pre-processing and argumentation as described in DARTS (Liu et al., 2019): zero padding the training images with 4 pixels on each side and then randomly cropping them back to $32 \times 32$ on CIFAR-10/100 and $64 \times 64$ on Tiny-ImageNet-200; randomly flipping training images horizontally; normalizing training images with the means and standard deviations along the channel dimension.

### A.2    SAMPLING OF RANDOM VARIANTS

For a $N$-node NAS cell, there are $\frac{(N-2)!}{(M-1)!}$ possible connections with $M$ input nodes and one output node. There are therefore hundreds to thousands of possible randomly connected variants for each popular NAS cell. The random variants of operations consist of a similar or even higher amount of architectures. Due to the prohibitive cost of comparing popular NAS cells with all variants, we randomly sample some variants to understand why the popular NAS cells are selected.

Given a NAS cell $C$, we fix the partial order of intermediate nodes and their accompanying operations. We then replace the source node of their associated operations by uniformly randomly sampling a node from their proceeding nodes in the same cell to get their randomly connected variants. Similarly, given a NAS cell $C$, we fix the partial order of intermediate nodes and their connection topologies. We then replace the operations couping each connection by uniformly randomly sampling from candidate operations to get their random variants of operations.

### A.3    ARCHITECTURES AND TRAINING DETAILS

For experiments on CIFAR-10/100 and Tiny-ImageNet-200, the neural network architectures are constructed by stacking $L = 20$ cells. Feature maps are down-sampled at the $L/3$-th and $2L/3$-th cell of the entire architecture with stride 2. For Tiny-ImageNet-200, the stride of the first convolutional layer is adapted to 2 to reduce the input resolution from $64 \times 64$ to $32 \times 32$. A more detailed building scheme can be found in DARTS (Liu et al., 2019).

In the default training setting, we apply stochastic gradient descent (SGD) with learning rate 0.025, momentum 0.9, weight decay $3 \times 10^{-4}$ and batch size 80 to train the models for 600 epochs on CIFAR10/100 and 300 epochs on Tiny-ImageNet-200 to ensure the convergence. The learning rate is gradually annealed to zero following the standard cosine annealing schedule. To compare the convergence under different learning rates in Section 4.1, we change the initial learning rate from 0.025 to 0.25 and 0.0025 respectively.

### A.4    REGULARIZATION

Since regularization mechanisms shall affect the convergence (Zhou et al., 2015), architectures are trained without regularization for a neat empirical study in Section 4. The regularization mechanisms are only used in Section 5 to get the converged generalization performance of the original and adapted NAS architectures on CIFAR-10/100 and Tiny-ImageNet-200 as shown in Table 1.

There are three adopted regularization mechanisms on CIFAR-10/100 and Tiny-ImageNet-200 in this paper: cutout (Devries & Taylor, 2017), auxiliary tower (Szegedy et al., 2015) and drop path (Larsson et al., 2017). We apply standard cutout regularization with cutout length 16. Moreover, the auxiliary tower is located at $2L/3$-th cell of the entire architecture with weight 0.4. We apply the same linearly-increased drop path schedule as in NASNet (Zoph et al., 2018) with the maximum probability of 0.2.

---

[5]Since no label is attached to Tiny-ImageNet-200 test dataset, we instead use its validation dataset to get the generalization performance of various architectures. We still name it as the test accuracy/error for brief.

## APPENDIX B    MORE RESULTS

### B.1    NAS ARCHITECTURES AND THEIR VARIANTS

We compare the width and depth of popular NAS architectures and their variants of random connections in Table 2. The random variants are sampled following the method in Appendix A.2. We further show the connection topologies of popular NAS and their partial random variants of connections in Figure 10 and Figure 11.

Table 2: Comparison of the width and depth of popular NAS cells and their randomly variants of connections. The name of the popular NAS cell is followed by its width and depth, which is separated by a comma. The width of a cell is conventionally computed by assuming that each intermediate node shares the same width $c$. Notably, the width and depth of random variants are in ascending and descending order respectively from $C_1$ to $C_{13}$. Moreover, the popular NAS architectures achieve the largest width and nearly the smallest depth among all the variants.

| Base Cell | $C_1$ | $C_2$ | $C_3$ | $C_4$ | $C_5$ | $C_6$ | $C_7$ | $C_8$ | $C_9$ | $C_{10}$ | $C_{11}$ | $C_{12}$ | $C_{13}$ |
|---|---|---|---|---|---|---|---|---|---|---|---|---|---|
| DARTS $(3.5c, 3)$ | $2c, 4$ | $2c, 4$ | $2c, 4$ | $2.5c, 4$ | $2.5c, 4$ | $2.5c, 3$ | $2.5c, 3$ | $2.5c, 3$ | $3c, 3$ | $3c, 3$ | $3c, 3$ | $3.5c, 3$ | $3.5c, 3$ |
| ENAS $(5c, 2)$ | $1.5c, 6$ | $1.5c, 5$ | $2c, 6$ | $2c, 6$ | $2.5c, 5$ | $2.5c, 5$ | $3c, 4$ | $3c, 3$ | $3.5c, 5$ | $3.5c, 4$ | $3.5c, 4$ | $3.5c, 3$ | $3.5c, 3$ |
| AmoebaNet $(4c, 4)$ | $1.5c, 6$ | $1.5c, 5$ | $1.5c, 5$ | $1.5c, 3$ | $2c, 6$ | $2c, 6$ | $2c, 4$ | $2.5c, 5$ | $2.5c, 3$ | $2.5c, 3$ | $3c, 3$ | $3.5c, 4$ | $3.5c, 3$ |
| NASNet $(5c, 2)$ | $1.5c, 6$ | $1.5c, 5$ | $2c, 6$ | $2c, 6$ | $2.5c, 5$ | $2.5c, 5$ | $3c, 4$ | $3c, 3$ | $3.5c, 5$ | $3.5c, 4$ | $3.5c, 4$ | $3.5c, 3$ | $3.5c, 3$ |

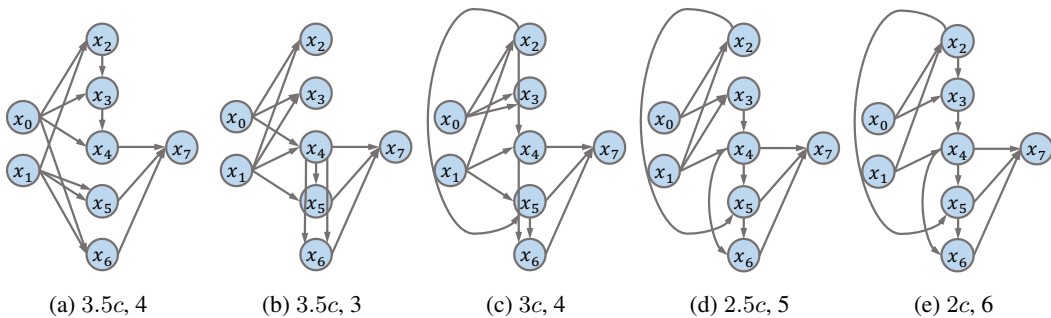

| (a) $3.5c, 4$ | (b) $3.5c, 3$ | (c) $3c, 4$ | (d) $2.5c, 5$ | (e) $2c, 6$ |

Figure 10: Connection topology of AmoebaNet cell (Real et al., 2019) and its part of randomly connected variants. Each sub-figure reports the width and depth of a cell separated by a comma. The leftmost one is the original connection from AmoebaNet normal cell and others are the ones randomly sampled. The width of a cell is also computed by assuming that each intermediate node shares the same width $c$. Notably, the original AmoebaNet cell has the largest width and almost the smallest depth among these cells.

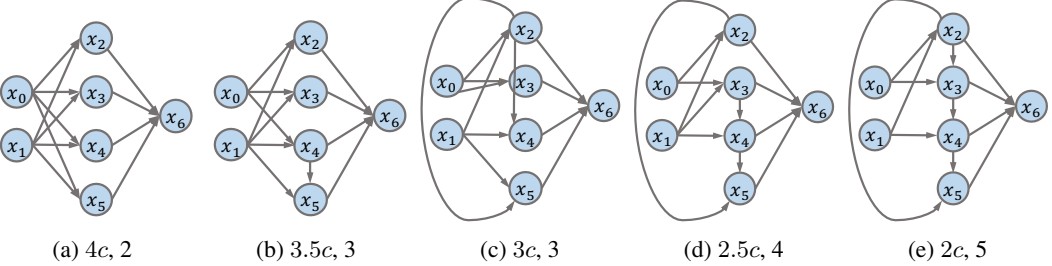

| (a) $4c, 2$ | (b) $3.5c, 3$ | (c) $3c, 3$ | (d) $2.5c, 4$ | (e) $2c, 5$ |

Figure 11: Connection topology of SNAS cell under mild constraint (Xie et al., 2019b) and its part of randomly connected variants. The width and depth of a cell are reported in the title of each plot. The leftmost one is the original connection from SNAS normal cell and others are the ones randomly sampled. The width of a cell is conventionally computed by assuming that each intermediate node shares the same width $c$. Notably, the original SNAS cell has the largest width and the smallest depth among these cells.

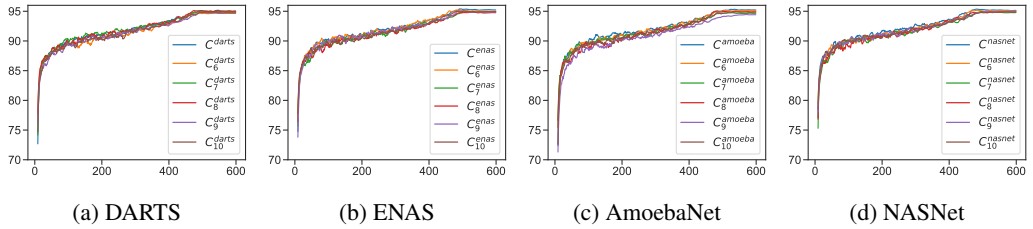

|     |     |     |     |
| --- | --- | --- | --- |
| (a) DARTS | (b) ENAS | (c) AmoebaNet | (d) NASNet |

Figure 12: More test accuracy (%) curves of DARTS, ENAS, AmoebaNet, NASNet and their random variants of operations on CIFAR-10 during training.

Table 3: Comparison of the parameter size (MB) of popular NAS cells and their randomly variants of operations. $C_0$ denotes the original NAS cell and $C_1$ to $C_{10}$ denote the random variants. Notably, there is a gap of $\sim 30\%$ between the parameter size of the smallest architecture and one of the largest architecture.

| Base cell | $C_0$ | $C_1$ | $C_2$ | $C_3$ | $C_4$ | $C_5$ | $C_6$ | $C_7$ | $C_8$ | $C_9$ | $C_{10}$ |
| --- | --- | --- | --- | --- | --- | --- | --- | --- | --- | --- | --- |
| DARTS | 3.35 | 3.37 | 2.84 | 2.70 | 2.98 | 3.19 | 2.43 | 3.49 | 2.88 | 3.31 | 2.81 |
| ENAS | 3.86 | 3.45 | 3.19 | 2.98 | 2.70 | 3.67 | 3.03 | 3.85 | 3.26 | 3.81 | 3.29 |
| AmoebaNet | 3.15 | 2.86 | 2.62 | 2.41 | 2.10 | 3.10 | 2.46 | 3.28 | 2.69 | 3.42 | 2.75 |
| NASNet | 3.83 | 3.45 | 3.19 | 2.98 | 2.70 | 3.67 | 3.03 | 3.85 | 3.26 | 3.81 | 3.29 |

## B.2 CONVERGENCE

In this section, we plot more test loss curves on CIFAR-10 (Krizhevsky et al., 2009) for original popular NAS architectures and their (12) randomly connected variants, as shown in Figure 13, Figure 14 and Figure 16. The depth and width of these 12 randomly connected variants can be found in Table 2. Notably, the width and depth of random variants (from $C_1$ to $C_{12}$) are in ascending and descending order respectively. Moreover, the popular NAS architectures achieve the largest width and nearly the smallest depth among all the variants. As shown in the following figures, the popular NAS cells, with larger width and smaller depth, typically achieve faster and more stable convergence than the random variants. Furthermore, with the increasing width and the decreasing depth, the convergence of random variants approaches to the original NAS architecture.

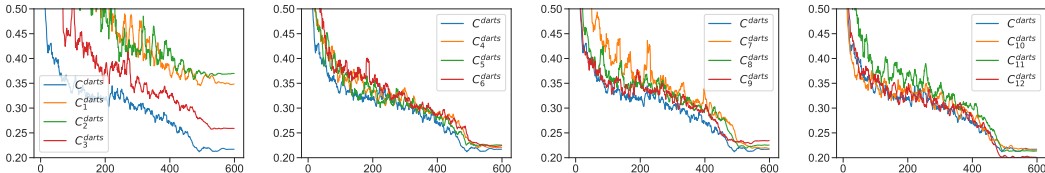

Figure 13: Test loss curves of DARTS and its variants on CIFAR-10 during training.

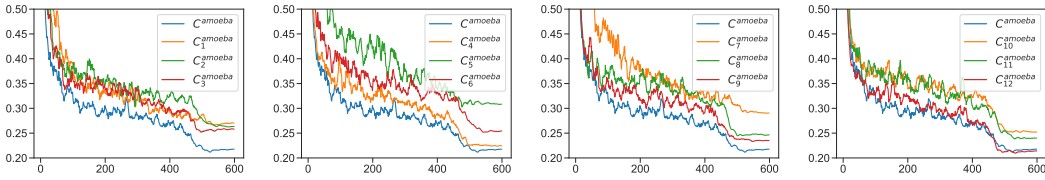

Figure 14: Test loss curves of AmoebaNet and its variants on CIFAR-10 during training.

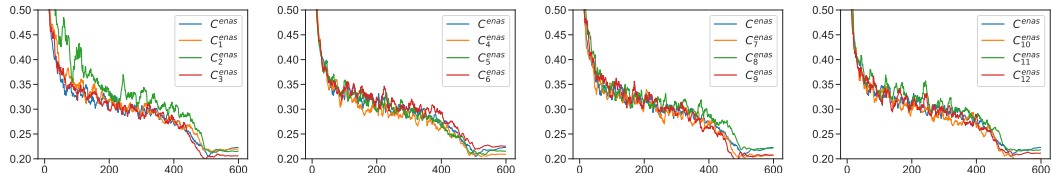

Figure 15: Test loss curves of ENAS and its variants on CIFAR-10 suring training.

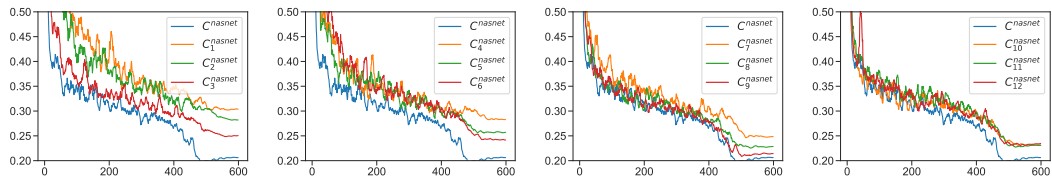

Figure 16: Test loss curves of NASNet and its variants on CIFAR-10 suring training.

### B.3 LOSS LANDSCAPE

In this section, we visualize loss landscapes for popular NAS architectures and their randomly connected variants. The depth and width of a cell are highly correlated. For example, the depth and width cannot reach their maximum simultaneously. With the increasing width, the average depth of cells grouped by the same width is decreasing as shown in Table 2. We therefore only group the results (including the ones from original NAS architectures) with various width levels of a cell for a better comparison. Notably, the architectures with wider and shallower cells have a smoother and benigner loss landscape, as shown in Figure 17, Figure 18, Figure 19 and Figure 20, which further supports the results in Section 4.2.1.

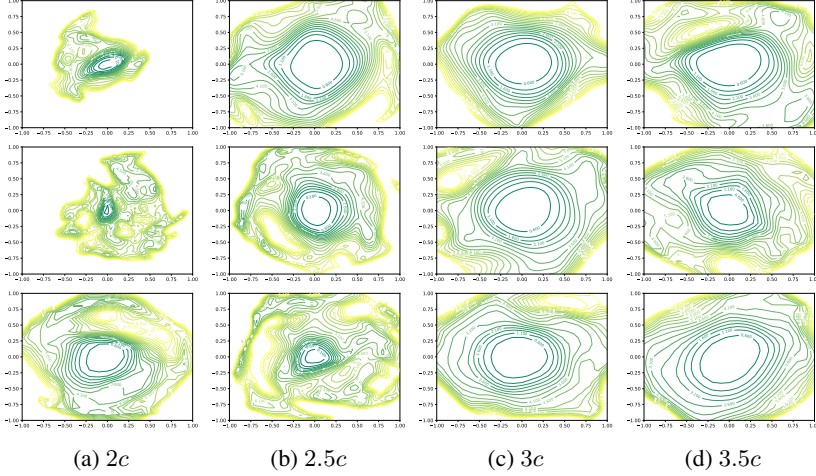

(a) $2c$      (b) $2.5c$      (c) $3c$      (d) $3.5c$

Figure 17: Loss contours of DARTS and its variants with random connections on the test dataset of CIFAR-10.

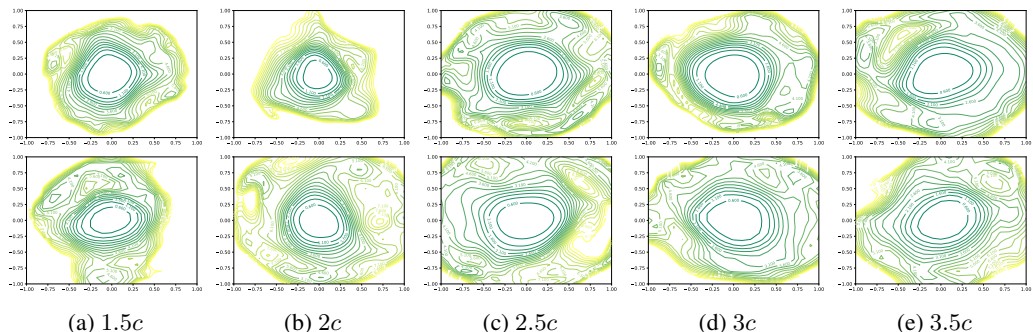

(a) $1.5c$     (b) $2c$     (c) $2.5c$     (d) $3c$     (e) $3.5c$

Figure 18: Loss contours of AmoebaNet and its randomly connected variants on the test dataset of CIFAR-10.

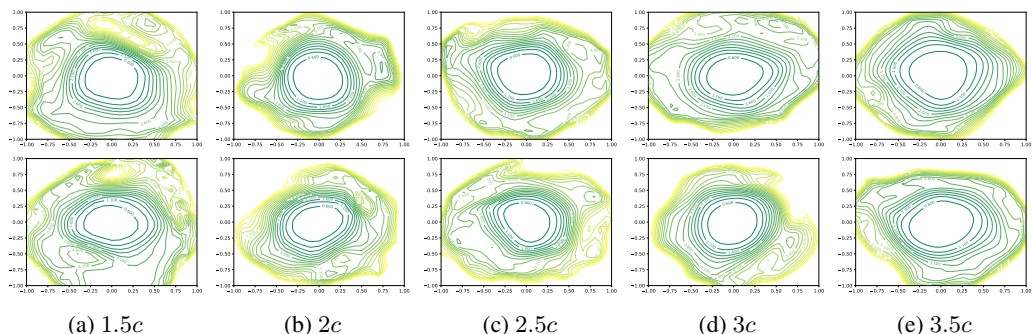

(a) $1.5c$     (b) $2c$     (c) $2.5c$     (d) $3c$     (e) $3.5c$

Figure 19: Loss contours of ENAS and its randomly connected variants on the test dataset of CIFAR-10.

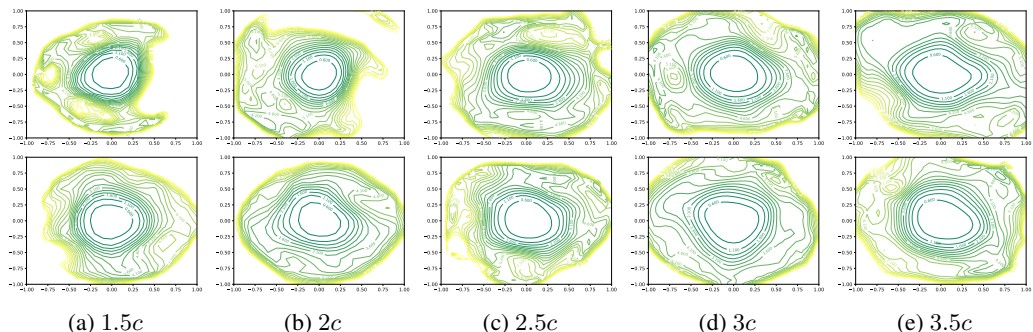

(a) $1.5c$     (b) $2c$     (c) $2.5c$     (d) $3c$     (e) $3.5c$

Figure 20: Loss contours of NASNet and its randomly connected variants on the test dataset of CIFAR-10.

## B.4 GRADIENT VARIANCE

In this section, we visualize the gradient variance (i.e., $g(\alpha, \beta)$ as defined in Section 4.2.2) for the popular NAS architectures as well as their variants with random connection, such as AmoebaNet in Figure 21, DARTS in Figure 22, ENAS in Figure 23 and NASNet in Figure 23. The $z$-axis has been scaled by $10^{-5}$ for a better visualization. Similarly, we group the results based on the width of cells. Notably, architectures with wider and shallower cells achieve relatively smaller gradient variance, which further confirms the results in Section 4.2.2.

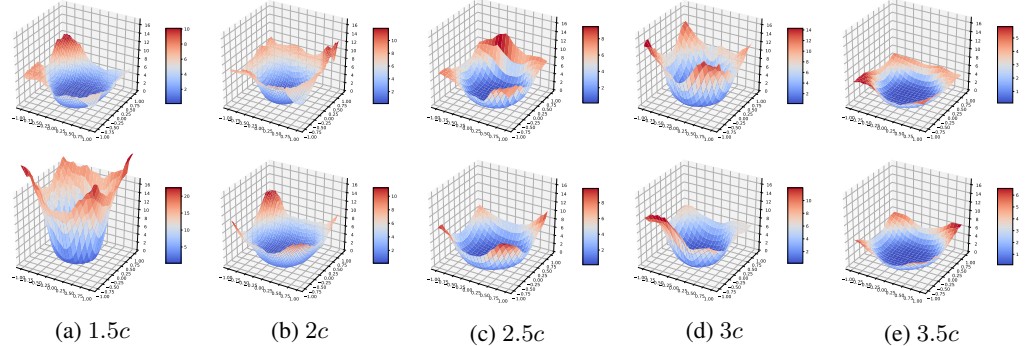

(a) $1.5c$     (b) $2c$     (c) $2.5c$     (d) $3c$     (e) $3.5c$

Figure 21: 3D surfaces of the gradient variance from AmoebaNet and its randomly connected variants on the test dataset of CIFAR-10.

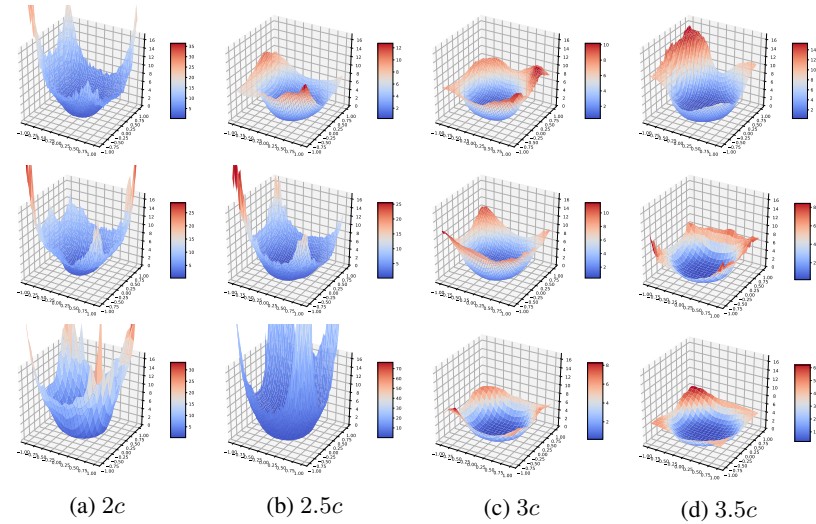

(a) $2c$     (b) $2.5c$     (c) $3c$     (d) $3.5c$

Figure 22: 3D surfaces of the gradient variance from DARTS and its randomly connected variants on the test dataset of CIFAR-10.

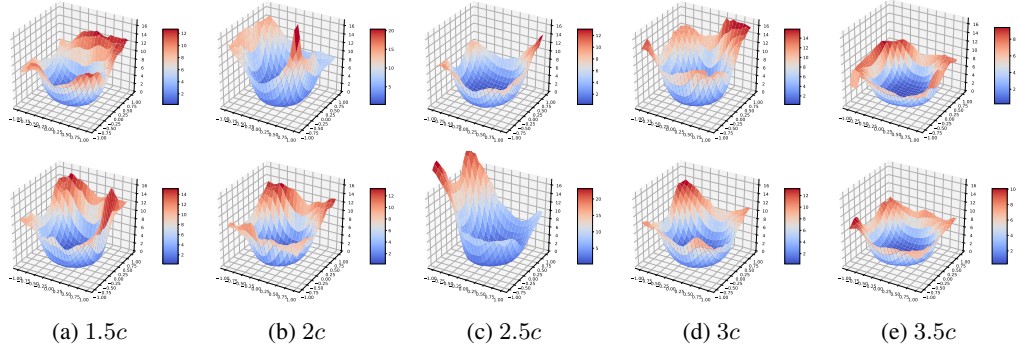

(a) $1.5c$     (b) $2c$     (c) $2.5c$     (d) $3c$     (e) $3.5c$

Figure 23: 3D surfaces of the gradient variance from ENAS and its randomly connected variants on the test dataset of CIFAR-10.

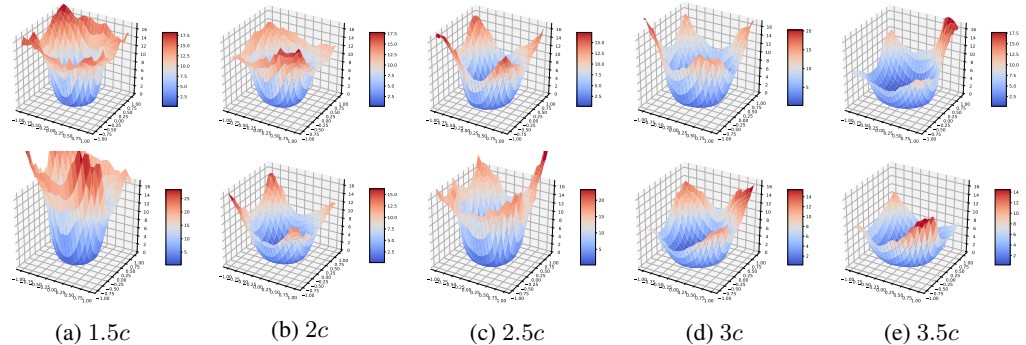

(a) $1.5c$     (b) $2c$     (c) $2.5c$     (d) $3c$     (e) $3.5c$

Figure 24: 3D surfaces of the gradient variance from NASNet and its randomly connected variants on the test dataset of CIFAR-10.

### B.5 ADAPTED TOPOLOGIES

In this section, we visualize the adapted architectures (in Figure 25) we investigate on in Section 5. Notably, The adapted connection topologies are not only applied in the normal cell but also the reduction cell. The adapted architectures are compared with popular NAS architectures to examine the impacts of the common connection pattern on generalization.

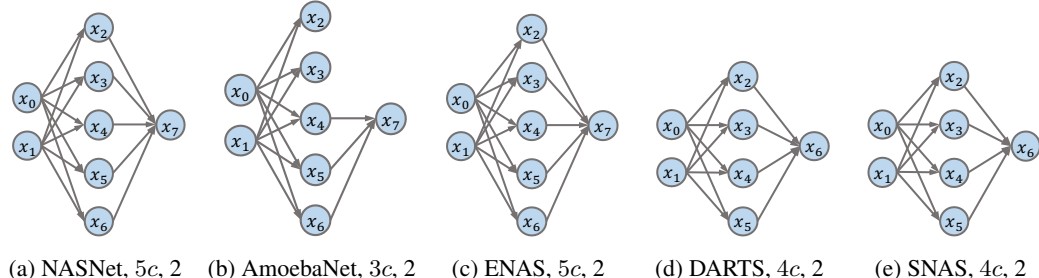

(a) NASNet, $5c$, 2    (b) AmoebaNet, $3c$, 2    (c) ENAS, $5c$, 2    (d) DARTS, $4c$, 2    (e) SNAS, $4c$, 2

Figure 25: Adapted topologies of cells from popular NAS architectures. The title of each sub-figure includes the name of the architecture, width and depth of the cell following our definition. Notably, these cells achieve the largest width and smallest depth in their original search space.

### APPENDIX C  THEORETICAL ANALYSIS

### C.1  SETUP

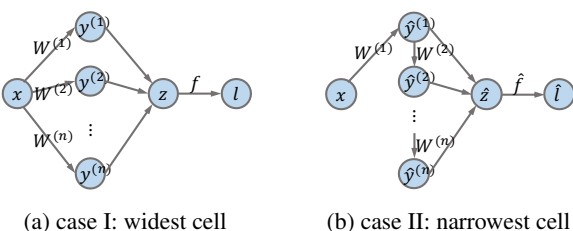

(a) case I: widest cell      (b) case II: narrowest cell

Figure 26: Two architectures to compare in the theoretical analysis: (a) architecture with widest cell; (b) architecture with narrowest cell. The notation $l$ and $\widehat{l}$ denote the values of objective function $f$ and $\widehat{f}$ evaluated at input $x$ respectively.

## C.2 Basics

We firstly compare the gradient of case I and case II shown in Figure 26. For case I, since $\boldsymbol{y}^{(i)} = W^{(i)}\boldsymbol{x}$, the gradient to each weight matrix $W^{(i)}$ is denoted by

$$\frac{\partial f}{\partial W^{(i)}} = \frac{\partial f}{\partial \boldsymbol{y}^{(i)}} \boldsymbol{x}^T \tag{1}$$

Similarly, since $\widehat{\boldsymbol{y}}^{(i)} = \prod_{k=1}^{i} W^{(k)}\boldsymbol{x}$ for the case II, the gradient to each weight matrix $W^{(i)}$ is denoted by

$$\frac{\partial \widehat{f}}{\partial W^{(i)}} = \sum_{k=i}^{n} (\prod_{j=i+1}^{k} W^{(j)})^T \frac{\partial \widehat{f}}{\partial \widehat{\boldsymbol{y}}^{(k)}} (\prod_{j=1}^{i-1} W^{(j)}\boldsymbol{x})^T \tag{2}$$

$$= \sum_{k=i}^{n} (\prod_{j=i+1}^{k} W^{(j)})^T \frac{\partial \widehat{f}}{\partial \widehat{\boldsymbol{y}}^{(k)}} \boldsymbol{x}^T (\prod_{j=1}^{i-1} W^{(j)})^T \tag{3}$$

$$= \sum_{k=i}^{n} (\prod_{j=i+1}^{k} W^{(j)})^T \frac{\partial f}{\partial W^{(k)}} (\prod_{j=1}^{i-1} W^{(j)})^T \tag{4}$$

Exploring the fact that $\frac{\partial \widehat{f}}{\partial \widehat{\boldsymbol{y}}^{(i)}} = \frac{\partial f}{\partial \boldsymbol{y}^{(i)}}$, we get (4) by inserting (1) into (3).

## C.3 Proof of Theorem 4.2

Due to the complexity of comparing the standard Lipschitz constant of the smoothness for these two cases, we instead investigate the block-wise Lipschitz constant (Beck & Tetruashvili, 2013). In other words, we evaluate the Lipschitz constant for each weight matrix $W^{(i)}$ while fixing all other matrices. Formally, we assume the block-wise Lipschitz smoothness of case I as

$$\left\| \frac{\partial f}{\partial W_1^{(i)}} - \frac{\partial f}{\partial W_2^{(i)}} \right\| \le L^{(i)} \left\| W_1^{(i)} - W_2^{(i)} \right\| \quad \forall\, W_1^{(i)}, W_2^{(i)} \tag{5}$$

The default matrix norm we adopted is 2-norm. And $W_1^{(i)}, W_2^{(i)}$ denote possible assignments for $W^{(i)}$.

Denoting that $\lambda^{(i)} = \left\| W^{(i)} \right\|$, which is the largest eigenvalue of matrix $W^{(i)}$, we can get the smoothness of case II as

$$\left\| \frac{\partial \widehat{f}}{\partial W_1^{(i)}} - \frac{\partial \widehat{f}}{\partial W_2^{(i)}} \right\| = \left\| \sum_{k=i}^{n} (\prod_{j=i+1}^{k} W^{(j)})^T (\frac{\partial f}{\partial W_1^{(k)}} - \frac{\partial f}{\partial W_2^{(k)}})(\prod_{j=1}^{i-1} W^{(j)})^T \right\| \tag{6}$$

$$\le \sum_{k=i}^{n} \left\| (\prod_{j=i+1}^{k} W^{(j)})^T (\frac{\partial f}{\partial W_1^{(k)}} - \frac{\partial f}{\partial W_2^{(k)}})(\prod_{j=1}^{i-1} W^{(j)})^T \right\| \tag{7}$$

$$\le \sum_{k=i}^{n} (\frac{1}{\lambda^{(i)}} \prod_{j=1}^{k} \lambda^{(j)}) L^{(k)} \left\| W_1^{(k)} - W_2^{(k)} \right\| \tag{8}$$

$$\le (\prod_{j=1}^{i-1} \lambda^{(j)}) L^{(i)} \left\| W_1^{(i)} - W_2^{(i)} \right\| \tag{9}$$

We get the equality in (6) since $j > i$ and $W^{(j)}$ keeps the same for the computation of block-wise Lipschitz constant of $W^{(i)}$. Based on the triangle inequality of norm, we get (7) from (6). We get (8) from (7) based on the inequality $\|WV\| \leq \|W\| \|V\|$ and the assumption of the smoothness for case I in (5). Finally, since we are evaluating the block-wise Lipschitz constant for $W^{(i)}$, $W_1^{(k)} = W_2^{(k)}$ while $k \neq i$, which leads to the final inequality (9).

### C.4 Proof of Theorem 4.3

Similarly, we assume the gradient variance of case I is bounded as

$$\mathbb{E} \left\| \frac{\partial f}{\partial W^{(i)}} - \mathbb{E} \frac{\partial f}{\partial W^{(i)}} \right\|^2 \leq (\sigma^{(i)})^2 \tag{10}$$

The gradient variance of case II is then bounded by

$$\mathbb{E} \left\| \frac{\partial \widehat{f}}{\partial W^{(i)}} - \mathbb{E} \frac{\partial \widehat{f}}{\partial W^{(i)}} \right\|^2 = \mathbb{E} \left\| \sum_{k=i}^{n} (\prod_{j=i+1}^{k} W^{(j)})^T (\frac{\partial f}{\partial W^{(k)}} - \mathbb{E} \frac{\partial f}{\partial W^{(k)}})(\prod_{j=1}^{i-1} W^{(j)})^T \right\|^2 \tag{11}$$

$$\leq n \mathbb{E} \sum_{k=i}^{n} \left\| (\prod_{j=i+1}^{k} W^{(j)})^T (\frac{\partial f}{\partial W^{(k)}} - \mathbb{E} \frac{\partial f}{\partial W^{(k)}})(\prod_{j=1}^{i-1} W^{(j)})^T \right\|^2 \tag{12}$$

$$\leq n \sum_{k=i}^{n} (\frac{\sigma^{(k)}}{\lambda^{(i)}} \prod_{j=1}^{k} \lambda^{(j)})^2 \tag{13}$$

We get (12) from (11) based on Cauchy-Schwarz inequality. Based on the inequality $\|WV\| \leq \|W\| \|V\|$ and the assumption of bounded gradient variance for case I in (10), we get the final inequality.

