# OpenReview forum: "Understanding Architectures Learnt by Cell-based Neural Architecture Search"
_ICLR.cc/2020/Conference — Accept (Poster)_

### Official Review · AnonReviewer1 · 2019-10-22
**Official Blind Review #1**

**Rating:** 3

**Review:**

Summary:

This paper tries to understand the characteristics of the architectures found by common NAS methods in the cell-search space. Specifically it characterizes the cell-search space used by DARTS, SNAS, AmeobaNet and finds that a most of these search methods find cells which are wide and shallow in depth (they give a specific definition of width and depth for characterizing cells). In fact these cells are usually the widest and shallowest architectures in their search space. The author empirically find that because these kinds of topologies converge faster during training and inevitably every NAS algorithm during search don't train upto convergence but only up to a bit and make decisions based on partially converged statistics there is a bias in selection towards these topologies. They also provide theoretical intuition to back-up these empirical findings.

Secondly they analyze the generalization performance of such wide and shallow cell structures accidentally emphasized by search procedures. They take the common cell structures found by common NAS algorithms (NASNet AmoebaNet, ENAS, DARTS, SNAS) and make them the widest and shallowest possible in the search space (following the SNAS cell connection pattern) while keeping number of parameters as constant as possible. They find that on cifar10 the test error of the adapted architectures usually increase a bit while on cifar100 the adapted architectures decrease a bit.

Comments:

- Overall the paper is interesting and well-written. Definitely liked the fact that wide and shallow networks are being accidentally biased towards during search. Liked the empirical analysis and theoretical insights backing it up.

- The generalization experiments suggest to me that on bigger datasets wider and shallower networks might be better for generalization actually. Can we take the cell architectures found by various algorithms and 'scale-up' to ImageNet by doing the usual trick of replicating more of the cells together and training? At least going by Table 1 I find myself not agreeing with the statement "The results above have shown that architectures with the common connection pattern may not generalize better despite of a faster convergence." On cifar100 wider and shallower is better. Perhaps on ImageNet they will be even better? So NAS algorithms' strategy of training partially may be exactly the right thing to do? Any thoughts?

- Any idea about if this pattern extends to RNN space as well or only limited to CNNs?

- Overall my main gripe is that while it is interesting findings but I am not sure I understood the main takeaway or significance of these results especially the generalization ones and how it informs search algorithm design.

**Experience Assessment:**

I have published one or two papers in this area.

**Review Assessment: Checking Correctness Of Derivations And Theory:**

I assessed the sensibility of the derivations and theory.

**Review Assessment: Checking Correctness Of Experiments:**

I carefully checked the experiments.

**Review Assessment: Thoroughness In Paper Reading:**

I read the paper at least twice and used my best judgement in assessing the paper.

---

> ### Author Response · Authors · 2019-11-08
> **Response to Reviewer #1**
>
> We thank you for the valuable comments and constructive questions. We respond to your questions as follows:
>
> 1.  Considering that training on ImageNet is very slow and we need to run many rounds of training for different cells, we may not be able to finish the experiments before the deadline. Therefore, we conducted extended experiments on Tiny-ImageNet-200. The results are shown below, which have been updated in the paper.
> ————————————————————--------------
> Architectures | original | adapted
> ————————————————————--------------
> NASNet       	  | 31.88    | 32.05
> AmoebaNet    | 32.22    | 33.16
> ENAS       	  | 30.68    | 31.36
> DARTS       	  | 30.58    | 31.33
> SNAS       	  | 32.40    | 32.61
> ————————————————————--------------
>  The results show that the widest and shallowest cells achieve worse generalization performance than the popular NAS cells. Given the better performance of adapted architectures on CIFAR-100 and worse performance on CIFAR-10 and Tiny-ImageNet-200, wider and shallower architectures are not guaranteed to achieve better generalization than narrower ones. Therefore, for better generalization performance, we still need to evolve/redesign NAS algorithms.
>
> 2. We only experiment on CNN since NASNet, SNAS, and AmoebaNet haven't conducted their experiments on RNN. Based on the results of DARTS and ENAS on RNN, it's not sufficient to get any insights.
>
> 3. The takeaway message for this paper is that current NAS algorithms choose architectures based on their convergence instead of the true generalization performance, which gives the bias to select wider and shallower cells. However, wider and shallower cells are not guaranteed to generalize better than other cells. This paper, therefore, helps us to understand what has been done by current NAS algorithms, why they do it in this way,  and why these algorithms may not be good enough. Thus, further design of NAS algorithms can focus more on how to precisely measure the generalization of candidate architectures to improve the performance of currently popular NAS architectures. Some work has started to think about the impacts of architecture itself while isolating the optimization process, such as [1].
>
> Reference
> [1] Gaier, A., & Ha, D. (2019). Weight Agnostic Neural Networks. arXiv preprint arXiv:1906.04358.

---

> > ### Comment · AnonReviewer1 · 2019-11-11
> > **Fair enough!**
> >
> > I agree that wide-shallow cells are not necessarily better at generalization. Thanks for the new experiments.

---

### Official Review · AnonReviewer3 · 2019-10-23
**Official Blind Review #3**

**Rating:** 6

**Review:**

--- Update after author's response ---

Thank authors for providing the detailed response to all my concerns. I am revising my rating to weak accept.

Specifically, generalising with 52 different variants of depth-width settings are enough, and the updated version is more solid than the earlier one. I think this paper should be accepted.


Summary:
The paper observed one common pattern of searched cell by 5 NAS algorithms, which is the cell found usually has large width but small depth structure, and claims the reason of such pattern is because architectures with shallow but wide structure converge fast during training, and thus are sampled by the NAS policy. To justify this fast convergence claim, the paper proposed to 1) define a width-depth level (width based on feature maps dimension, and depth based on its DAG connection) for each cell in the search space, and randomly sampled one architecture at each level on top of some best-cell discovered by NAS algorithms, 2) training them on original task from scratch independently, and provide visualization of training curve under various learning rate settings, loss landscape as well as gradient variance plot. For theoretical analysis, the paper formulates the narrowest and widest cells, and showing the difference of gradient is bounded by its Lipschitz smoothness of parameter matrices, and usually such variance indicates the widest architecture could converge faster than the narrowest one.

Whilst this observation is interesting, I found the empirical and theoretical justifications seem to be insufficient to support the claims for the following reasons, 1) The experiments are overwhelmingly built on top of **1** architecture (even obtained from random sampling) of each width-depth level, and it may not well represent the common behavior in the search space, thus the generalization of these claims remains questionable; 2) Theoretical analysis showing the gradient variance difference of narrowest architecture is bounded comparing to the widest cell, however, in practice, these architectures are not properly evaluated. Without proper extension, it is hard to conclude that such difference bound between a wider and narrower architecture pair; 3) experiments in supplementary negatively affects the generalizability of this paper, since the observed trend on DARTS search space does not agree with the one on AmoebaNet and SNAS cases; 4) paper claims the NAS algorithm tends to pick the fast converging architectures more than those late converged one, while intuitive, without showing some detailed process about how NAS algorithms converge, and which architectures they actually sampled during the search phase.

Nevertheless, I do agree that this paper has a clear motivation, and the observation is interesting and important. If the author could show the conclusion still hold after scaling the experiments, I am learning to accept this paper in the end.


Strength
+ Observation of these NAS algorithms tends to pick wide but shallow cell type is interesting, motivation of this work is well justified.
+ Experiments are throughout, instructive under the paper's current setting.
+ Theoretical justification for the gradient variance between the narrowest and widest cells are sound.
+ Paper is well written and easy to follow, the figures are presented clearly.

Weakness

- Insufficient experimental setting to support the claim.
My major concern is that under the current experiment design, it is not clear if the observation is well justified, and impedes the main contribution. As mentioned in the review's summary, it is not that convincing that, for each width-depth level, one architecture is enough. I understand exploiting all the variants is resource consuming, however, without such experiments, the current experiments can be impacted by many factors, such as, 1) as in Appendix A2, for each level, the paper "fixed the partial order of their intermediate nodes" and "replace the source node of their associated operations by uniformly randomly sampling a node from their proceeding nodes in the same cell", if I understand correctly, this means the operations will remain the same. However, since these best architectures are searched over both operation and topology connection, new architectures generated from this way may be sub-optimal, hence the larger gradients or slow convergence is not only because they are "narrower" and "deeper". Without proper isolation, it is not possible to conclude as in paper.

To this end, I suggest author provide the following experiments, 1) sampling all the variants at each level, within a search space like NASBench-101[1], where all the architecture performances are known, 2) at least sampling a sufficient number in the current space (probably > 30 to be statistically significant); 3) Random sampling a small topological variances, then run NAS search algorithms to search the best operation set, then redo the experiments in the paper.

- Trending of DARTS evaluation results does not agree with SNAS and AmoebaNet.
In Figure  5 (loss landscape plot) and Figure 6 (gradient variance heatmap) for DARTS, the wider-shallower architectures are better comparing to narrower-deeper ones, however, this trend is not significant in Figure 14,16 for AmoebaNet space and Figure 15, 17 for SNAS space. After a closer look, I noticed in Figure 2, the Darts C3 and C4, input node x0 is not connected to the graph, while C, C1, C2, and all the topologies in Figure 10, 11, x0 is connected. Could the author(s) comment on this? Will this be the reason why the C3, C4 DARTS are worse than other architectures?

Minor comments
Page 1 - Introduction paragraph 3 line 1 - 'typologies': is this referring to 'topologies'?
Table 1 - Adapted architectures on CIFAR-10 are mostly worse, even on CIFAR-100, they are better in a small margin. This does not support the claim well.
Figure 9 - Could the author provide the width-depth information for each index?

Reference
[1] Ying et al. NAS-Bench-101: Towards Reproducible Neural Architecture Search, ICML'19.

**Experience Assessment:**

I have published one or two papers in this area.

**Review Assessment: Checking Correctness Of Derivations And Theory:**

I carefully checked the derivations and theory.

**Review Assessment: Checking Correctness Of Experiments:**

I carefully checked the experiments.

**Review Assessment: Thoroughness In Paper Reading:**

I read the paper thoroughly.

---

> ### Author Response · Authors · 2019-11-11
> **Response to Reviewer #3 (Part 2)**
>
>
> **Question**
> Table 1 - Adapted architectures on CIFAR-10 are mostly worse, even on CIFAR-100, they are better in a small margin. This does not support the claim well.
> **Response**
> Popular NAS architectures are already very wide as shown in Fig. 1, and thus, the width of adapted cells can only increase marginally. Consequently, a small performance difference is expected. However, these results are still reasonable to support our claim. Moreover, Fig.9 further supports our claim with more significant and consistent results.
>
> **Question**
> - Page 1 - Introduction paragraph 3 line 1 - 'typologies': is this referring to 'topologies'?
> - Figure 9 - Could the author provide the width-depth information for each index?
> **Response**
> We have corrected the typo and update the width-depth information of Fig. 9 in Table 2 of Appendix B.1.
>
> **Question**:
> -paper claims that the NAS algorithm tends to pick fast converging architectures more than those late converged one, while intuitive, without showing some detailed process about how NAS algorithms converge, and which architectures they actually sampled during the search phase.
> **Response**:
> We do not show the adaption of NAS architectures and convergence of NAS algorithms during the search process because the final selected architectures (the ‘optimal’ choice) can already reveal the tendency of NAS algorithms. Moreover, the architectures sampled during the search phase are not necessary to reveal the tendency of NAS algorithms since it may approach the local optimal and thus may have totally different tendencies compared to the final optimal choice (‘global’ optimal). Therefore, we mainly investigate the difference between these selected architectures and other candidate architectures. While these selected NAS architectures indeed converge faster as shown in Fig. 3,4,13,14,15, we therefore achieve the claim that NAS algorithms tend to select fast converging architectures.
>
> **Question**:
> -Theoretical analysis showing the gradient variance difference of narrowest architecture is bounded comparing to the widest cell, however, in practice, these architectures are not properly evaluated. Without proper extension, it is hard to conclude that such difference bound between a wider and narrower architecture pair.
> **Response**:
> We agree with you that theoretical results can only explain some simplified cases. However, the understanding based on simplified cases is still of great importance to understand how the connection may impact the properties (smoothness and gradient variance) NAS architectures in more complex cases. To further support our conclusion, we have also provided empirical results in this paper regarding smoothness and gradient variance as shown in Appendix B.3 and B.4.

---

> ### Author Response · Authors · 2019-11-11
> **Response to Reviewer #3 (Part 1)**
>
>
> **Questions**
> - However, since these best architectures are searched over both operation and topology connection, new architectures generated from this way may be sub-optimal, hence the larger gradients or slow convergence is not only because they are "narrower" and "deeper". Without proper isolation, it is not possible to conclude as in paper.
> - (Suggestion 3) Random sampling a small topological variances, then run NAS search algorithms to search the best operation set, then redo the experiments in the paper.
> **Response**
> We agree with you on the possible impacts of sub-optimal operations on the convergence and generalization. The suggestion of applying NAS search algorithms to search the best operations for the randomly connected variants would help isolate the impacts of sub-optimal operations. Alternatively, we have sampled and trained random variants of operations for popular NAS architectures while fixing the connection topologies. We find that the convergence for random variants of operations remains nearly the same as their original NAS architectures as shown in Fig. 5, 12. This finding shows that the sub-optimal (even random) operations have a limited impact on the convergence, which is quite enlightening. Consequently, we can isolate the impacts of operations on convergence. The investigation on the relation between connections and convergence as conducted in the paper is still valid.
>
> **Question**
> - As mentioned in the review's summary, it is not that convincing that, for each width-depth level, one architecture is enough.
> - (Suggestion 1) Sampling all the variants at each level, within a search space like NASBench-101[1], where all the architecture performances are known.
> - (Suggestion 2) At least sampling a sufficient number in the current space (probably > 30 to be statistically significant).
> **Response**
> Yes. We agree with you that with more variants, the results would be more reliable.
> We have updated more results of convergence, loss landscape and gradient variance in Appendix B.2, B.3 and B.4 respectively for various width-level. These results further support our claim. In fact, given a fixed number of nodes in the cell, the depth and width are highly correlated. For example, the depth and width cannot reach their maximum simultaneously. We therefore group these results by various width levels as shown in Fig. 16-23. We have also updated the results with more variants in Fig. 9 and Fig. 13-15. There are in total 52 variants of different width and depth as shown in Table 2.
>
> **Question**
> - Trending of DARTS evaluation results does not agree with SNAS and AmoebaNet.
> **Response**
> Trending of DARTS evaluation results actually is consistent with SNAS and AmoebaNet. Further explanations over the node x0 should help address the concern over the discrepancy of the results between different architectures. The isolated x0 is partially responsible for the chaotic loss landscape and the larger gradient variance for C3 and C4 DARTS. The node x0 is actually the implicit shortcut among cells. While the number of cells is fixing, the shortcut can change the sequential topology of cells and enlarge the number of cells on the same level (super-layer). Therefore, shortcut helps widen and shorten architectures. As shown in [1], adding shortcuts can help smooth the loss landscape of ResNet. In contrast, removing shortcut (isolating x0 in this paper) will impair the smoothness of the loss landscape. This result is mainly based on the width of the whole architecture, whereas our claim about the benefits of wider and shallower cells for smoothness is based on the width of a cell. However, these two results are consistent regarding the impacts of width and depth. Moreover, for DARTS and other NAS cells, more random variants on various width-depth levels have been sampled and trained. Their results are shown in Appendix B.2, B.3 and B.4 . We omit the results for SNAS since half of operations in a SNAS cell are skip connections. The skip connections make the cell wide and shallow implicitly and their results are therefore less significant. The results in Appendix B.2, B.3 and B.4 reveal that while there are no isolating input nodes, larger width still leads to smoother loss landscape and smaller gradient variance, which is consistent with our claim.
>
> Reference
> [1] Li, H., Xu, Z., Taylor, G., Studer, C., & Goldstein, T. (2018). Visualizing the loss landscape of neural nets. In Advances in Neural Information Processing Systems (pp. 6389-6399).

---

### Official Review · AnonReviewer2 · 2019-11-01
**Official Blind Review #2**

**Rating:** 8

**Review:**

The paper makes an interesting observation and tries to explain what causes it: architecture search methods tend to favor models that are easier to optimize, but not necessarily better at generalization.  I lean towards accepting the paper but there is some clear room for improvement.

The paper shows that NAS methods comes up with shallower but wider cells.
- These cells are easier to optimize because they have a smoother loss surface and lower gradient variance.
- Being easy/fast to optimize is favored by a NAS method because the models are typically not trained to convergence. Instead they are evaluated after a brief period of training.
- This leaves the question: why are they smoother.

Before going into the comments I want to state that I am very happy to see that a paper providing an analysis of existing methods to enhance our understanding. This is very much needed in the architecture search community.

==== Comments ====
- Could you describe what exactly is plotted in Fig. 5, 6 and 7. Specifically what are the aces. . It would make the manuscript more self contained. This point is also one of the main reasons why I did not give the manuscript a higher score. I am unsure of what is plotted but I am giving this manuscript the benefit of the doubt because it is consistent with my own experience.

- So far in the main text, most results focus on DARTS. It would be interesting to see the same consistent behavior is also present when comparing cells originating from different search spaces. (This is slightly different than the setting where the experiment is repeated on isolated search spaces). If this comment is unclear, please ask for a clarification.

- Would it be possible to alter the conclusions by modifying the initialization of the weight matrix. It appears to be the case that the smoothness and the variance both depend on the eigenvalues of the weight matrices. If we could make them more well behaved we could potentially make the narrower architectures train faster?

- Can you use a better term than common connection pattern in the abstract and conclusion. In general the abstract and conclusion could be written in a crisper and more to the point way.

- Please update Table 1 to actually include parameter sizes. This would make the result more reliable. Also the adapted cells need to be explicitly provided.

- Could you provide me with a better understanding of the difference between standard lipschitz and block lipschitz.

==== More minor issues ====
#) Correction required
In section 2, Zoph et al. is cited where it is argued that weight sharing could detrimental for the performance of nas methods. However this paper does not use weight sharing

#) Corrections recommended
Xie et al. 2019 is said to be state of the art. In the actual publication the authors only claim to be competitive. I believe that this is a more balanced statement since those results are not always best on all dimensions (sometimes better on FLOPS but not parameters, unclear whether they would be faster on an actual device, on the large scale results they do lag behind in quality too.)

#) My apologies for the slight digression but I do not think that using the Sciuto et al. paper is a good reference to discount weight sharing approaches or claim that random search is equally good. Some of their experiments are performed on a tiny search space with only 32 models. This gives random search a high probability of getting the right model, while a simple bias in the algorithm might cause it to be sub-optimal. That experiment does not show that random search is as good. Also on the PTB task, their reported results are worse than the open sourced implementation for DARTS. This means to me that this paper cannot be used as a reliable reference.


**Experience Assessment:**

I have published in this field for several years.

**Review Assessment: Checking Correctness Of Derivations And Theory:**

I assessed the sensibility of the derivations and theory.

**Review Assessment: Checking Correctness Of Experiments:**

I carefully checked the experiments.

**Review Assessment: Thoroughness In Paper Reading:**

I read the paper thoroughly.

---

> ### Author Response · Authors · 2019-11-08
> **Response to Reviewer #2**
>
> We thank you for the valuable comments and constructive suggestions. We respond to your questions as follows:
>
> 1. The x-axis and y-axis for Fig. 5, 6 and 7 are $\alpha$ and $\beta$ respectively, which are the step sizes to permute the trained parameters $w^*$ along sampled direction $w_1$ and $w_2$ respectively. We plot loss landscape $s(\alpha, \beta)$ (defined in Sec. 4.2.1) in Fig5, represented as 2-D contour. We plot gradient variance $g(\alpha, \beta)$ (defined in Sec. 4.2.2) as heat map and 3D surface respectively in Fig. 6 and Fig. 7. All these plots are to visualize and estimate the properties (smoothness and gradient variance) of the objective function with parameters around their optimal value in a low dimension space. We have updated the explanation for $\alpha$ and $\beta$ in the first paragraph of Sec.4.2.1.
>
> 2. We have repeated experiments on various NAS cells (e.g., DARTS, AmoebaNet, ENAS), which share similar but slightly different search spaces. We choose these search spaces for experiments because they are widely adopted by other NAS papers. The results are mainly in the Appendix and are consistent with the results in the main text. More results of convergence, loss landscape and gradient variance have been updated in Appendix B.2, B.3 and B.4 respectively for various search spaces (i.e., DARTS, AmoebaNet, ENAS, NASNet).  Hope this reply will address your concern on the search space.
>
> 3. The conclusion is more likely to stay the same, although we alter the initialization methods. The initialization can impact the convergence as shown in Theorem 4.1 in our paper. However, for those narrower and deeper cells, a good initialization (near local optimal parameters) is hard to get because of their chaotic loss landscape as shown in Figure 5. Besides, the Theorem 4.2 and 4.3 show that smoothness and gradient variance of the narrowest cell are impacted by the eigenvalue of multiple weight matrices. It's hard to keep most of the weight matrices well behaved during training and therefore it is hard to make those narrower cells smoother than the wider ones. Thus, wider cells are still more likely to converge faster than narrower cells under different initialization methods.
>
> 4.
> (1) We have updated the wrong citation (Zoph et al.) for weight sharing.
> (2) We have updated the performance of Xie et al. from 'state of the art' to 'competitive'.
> (3) We have substituted most of the common connection pattern (including the one in abstract and conclusion) into cell width and depth for a better understanding while maintaining the context. However, we keep this term for the Observation 3.1 since it’s a common pattern which shows by various popular NAS architectures. We have re-written the abstract and conclusion to make them more concise and to the point.
> (5) We have updated the parameter size of Table 1 for the comparison. Moreover we have updated the results on Tiny-ImageNet-200 in Table 1 to further support our claim on larger dataset.
>
> 5. Given the function $f(x)$, the standard Lipschitz smoothness is defined on the whole coordinates of $x$:
> $\|\frac{\partial f}{\partial x_1} - \frac{\partial f}{\partial x_2}\| \leq L\|x_1 - x_2\|$
> Here the $x_1$ and $x_2$ are the all possible assignments of whole $x$.
> However, the definition of block-wise or block coordinate-wise Lipschitz smoothness is defined on separated sets of coordinates of $x$. Particularly, the block-wise Lipschitz smoothness for $i_{th}$ coordinate of $x$ can be defined as
> $\|\frac{\partial f}{\partial x_1^{(i)}} - \frac{\partial f}{\partial x_2^{(i)}}\| \leq L_i\|x_1^{(i)} - x_2^{(i)}\|$
> Here the $x_1^{(i)}$ and $x_2^{(i)}$ are the all possible assignments of $i_{th}$ coordinate of $x$ while keeping the other coordinates the same.
> As stated in [1], $f(x)$ is also Lipschitz smoothness by only assuming its block-wise Lipschitz smoothness.
>
>
> Reference
> [1] Beck, A., & Tetruashvili, L. (2013). On the convergence of block coordinate descent type methods. SIAM journal on Optimization, 23(4), 2037-2060.

---

### Author Response · Authors · 2019-11-13
**Summary of updates in the paper**

Dear reviewers and all,

We have updated our manuscript based on the reviewers’ comments and suggestions. Our paper has been greatly improved thanks to the reviewers’ valuable comments and constructive suggestions. We hope the reviewers may check the updated version and we welcome any further suggestions or feedback.

To summarize our main changes:

1. We have updated the abstract and conclusion, and updated ‘common connection pattern’ with ‘cell width and depth’ following the suggestion of Reviewer #2.

2. We have updated the explanation for the axises of the loss landscape and gradient variance plots in Sec. 4.2.1 following the comments of Reviewer #2.

2. We have sampled and trained more randomly connected variants. We have provided more results of their generalization, parameter size, convergence, loss landscape and gradient variance for various width levels in Fig. 9, Appendix B.1, B2, B3, and B4 respectively following the suggestion of Reviewer #2 and #3.

3. We have sampled and trained variants of random operations for popular NAS architectures. We have provided their results in Fig. 5 and 12 including their parameter size in Table 3 following the comments of Reviewer #3.

4. We have updated Table 1  with the results of Tiny-ImageNet-200 following the comments of Reviewer #1.

5. We have updated mirror typos or errors following the suggestion of Reviewer #2 and #3.

6. We have updated the response to Reviewer #1, #2 and #3 accordingly based on the revised manuscript.

Kind regards,
Authors

---

### Decision · Program_Chairs · 2019-12-19

**Decision:**

Accept (Poster)

**Comment:**

The paper reports interesting NAS patterns, supported by empirical and theoretical evidence that the pattern arises due to smooth loss landscape. Reviewers generally agree the this paper would be of interest for the NAS researchers. Some questions raised by reviewers are answered by authors with a few more extra experiments. We highly recommend authors to carefully reflect on reviewers both pros and cons of the paper before camera ready.